# A New Collector for Effectively Increasing Recovery in Copper Oxide Ore-Staged Flotation

**Renfeng Zhu \*, Guohua Gu, Zhixiang Chen, Yanhong Wang and Siyu Song**

School of Minerals Processing and Bio-Engineering, Central South University, Changsha 410083, China; guguohua@163.com (G.G.); chenzhixiang93@163.com (Z.C.); yanhong.wang@csu.edu.cn (Y.W.); 165612115@csu.edu.cn (S.S.)

**\*** Correspondence: nicky-xiaozhu@163.com; Tel.: +86-0731-8883-0960

**Abstract:** A new method, staged flotation for effectively increasing the recovery of ultra-fine copper oxide ore with a new type of collector (ZH-1, C3-5 carbon chain xanthate) is proposed for the first time. The flotation process and mechanism were examined by flotation tests, entrainment rate analysis, laser particle size experiments and microscopic imagery as well as economic feasibility analysis. It was demonstrated that the collector isoamyl sodium xanthate (ISX) shows a good collection ability (recovery exceeded 95%) for azurite, but the recovery was relatively much lower for malachite (only near 80%) due to the different particle size distribution. The new type of xanthate ZH-1 has shown a high-efficiency collection performance for fine-grained malachite. The recovery achieved for −10 μm malachite was more than 95% when the ZH-1 dosage was 150 mg/L, while the average particle size of −10 μm malachite sharply increased from 4.641 μm to 9.631 μm. The batch flotation results indicated that the copper oxide flotation recovery increased from 79.67% to 83.38%, and the grade also raised from 18.08% to 18.14% after using the staged flotation technology with ZH-1 as collector during the flotation of −25 μm ore. It was confirmed that this technology was quite effective for the recovery of copper oxide at the Dishui Copper Processing Plant, which successfully increased its gross profit by 1.6 million US$ per year.

**Keywords:** staged flotation; malachite; azurite; fine-grained; mixed xanthate; flocculation

## 1. Introduction

In order to improve the utilization rate of copper oxide ore resources, it is necessary to finely grind it for achieving liberation of as much as possible due to the fine inlay size of the valuable minerals, including malachite and azurite [1–6]. However, as particle size decreases, the flotation behavior of copper oxide mineral particles undergoes many changes. For instance, the fine-grained copper oxide ore is poorly floatable and displays a lower recovery during the flotation process [7]. Up to now, sulfide flotation is still one of the most widely used processes in copper oxide ore dressing, but it is unable to solve the problem of low recovery of micro-fine oxidized copper ores [8,9]. According to statistics, about one sixth of the copper resource is lost to tailing ascribed to fine particles that cannot be effectively recovered [10,11]. Therefore, how to improve the flotation vulcanization efficiency of fine-grained copper oxide ore has an important role in meeting the market's growing demand for copper, and it is also significant for the conservation of mineral resources.

The Dishui Copper Processing Plant is located in Qarqi Town (Aksu Prefecture, Xinjiang Province, China). The mining area is about 21 square kilometers, with the copper grade more than 1%, and the production scale is 3000 tons per day. The embedded granularity of the useful minerals, such as malachite and azurite, is not uniform in the copper ore. Therefore, it is difficult to determine the best grinding fines in a section of grinding since too much fineness will generate over-grinding which

eventually worsen the flotation process, but too low fineness will cause part of the copper minerals to be difficult to dissociate completely, and undissociated minerals cannot be recovered by flotation. At present, the on-site control grinding fineness of Dishui Copper Processing Plant is about 90%. The multi-sampled and sieved grinding products show that the +74 µm copper grade is the highest, generally above 0.30%, but the −25 µm slime content is more than 60%, which not only causes a waste of the flotation reagent, but also deteriorates flotation indicators. The fine-grained particles possess three main characteristics, including small mass, large specific surface area and high surface energy [12–14]. It is difficult to overcome the energy barriers between the ore particles and the bubbles then adhere to the surface of the bubbles. Besides, the large specific surface area and high surface energy cause severe non-selective agglomeration between the gangue particles and the valuable particles, resulting in a larger amount of reagent consumption during the flotation process and a decreasing in the selectivity of reagent adsorption [14–17].

Against this background, we proposed that the idea of stage treatment in the beginning, which including employing leaching or combination of metallurgy to dispose the −25 µm slime that separated from the grinding product by screening, and other −74 ~ +25 µm particles were treated by a conventional flotation process. In this way, the consumption of reagents can be greatly reduced during the flotation of +25 µm ore, especially the amount of xanthate collector. In our follow-up research, we conducted an acid leaching and ammonia leaching experiment on −25 µm products. However, the results showed that there are still many problems that cannot be solved in the application of this method. For example, consumption of a large amount of acid during the acid leaching process, the separation problems between solid and liquid and lowing leaching rate (only about 75%). On the other hand, the ammonia leaching effect is better than that of acid leaching, and the recovery of copper is about 85%. However, it also has the same problem of large consumption of chemicals, more seriously, ammonia volatilization during the leaching process will have a great harmful impact on the environment protection. Part of −25 µm slime could not be effectively treated and was stranded for a long time. Chen et al. [16] have shown that it is possible to increase the size of the fine particles by performing flocculation between particles, which enables the effective recovery of fine-grained minerals. The selective flocculation process was to add a polymer collector to the slurry after slurry adjustment, and selectively flocculated the target mineral under the action of shearing field, and then used flotation, re-election or de-sludge processes to achieve the separation of target minerals.

In this paper, we have designed a new type of collector, ZH-1, with excellent flocculation selectivity and collecting capacity for −25 µm valuable minerals, and finally improved the flotation recovery for this fraction of malachite. The mechanism of action between ZH-1 and mineral was studied by flotation kinetics, entrainment rate calculation, laser particle size analysis and microscopic imaging analysis. In addition, locked-cycle tests of grading flotation and economic feasibility calculation were carried out in this research, aiming to further verify the feasibility of the technical scheme and increase the recovery of copper oxide, meanwhile improving the utilization of mineral resources.

## 2. Materials and Methods

### 2.1. Materials

Pure malachite, azurite and industrial samples were obtained from the Dishui Copper Processing Plant, Qarqi, Xingjiang, China. The purity of the mineral sample was more than 90% by X-ray diffraction analysis, as shown in Figure 1. The chemical analysis results of the practical sample were listed in Table 1. Hand-selected crystals of these minerals were crushed to −1 mm in a laboratory roll crusher. The crushed products were then ground in a ceramic ball mill and the product was screened into three narrow size fractions: −38 + 25 µm, −25 + 10 µm, and −10 µm. The size distribution of narrow size fractions is summarized in Table 2.

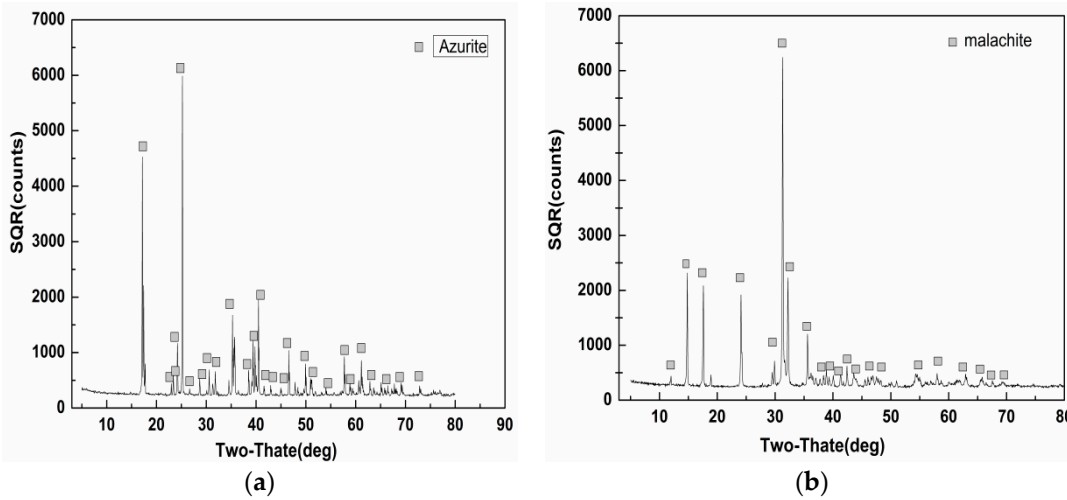

**Figure 1.** X-ray diffraction (XRD) spectra of pure (**a**) Azurite and (**b**) Malachite for tests.

**Table 1.** Chemical analysis results of practical samples.

| Element | Azurite | Malachite | Chalcocite | Limonite | Ilmenite | Pyrite | Rutile |
|---|---|---|---|---|---|---|---|
| Content/% | 0.72 | 0.59 | 0.22 | 0.73 | 0.35 | 0.30 | 0.25 |
| Element | Calcite | quartz | plagioclase | muscovite | chlorite | dolomite | others |
| Content/% | 24.93 | 22.83 | 15.25 | 9.88 | 9.51 | 7.53 | 6.91 |

**Table 2.** The size distribution of narrow size fractions.

| Size Fraction/μm | $d_{90}$/μm | Mean Size/μm |
|---|---|---|
| −38 + 25 | 34.876 | 33.213 |
| −25 + 10 | 23.454 | 22.647 |
| −10 | 8.535 | 6.375 |

The new type of collector, ZH-1, is based on mixed xanthates (3–5 carbon chain lengths) in a certain ratio. It was supplied by the Institute of Multipurpose Utilization of Mineral Resources Flotation Reagents Limited Company, Guangzhou, China. The solution was made up to the required volume by adding distilled water and then left to equilibrate overnight. Fresh solutions were prepared every three days. In the flotation tests, ZH-1 or isoamyl sodium xanthate (ISX) or isopentyl potassium xanthate (IPX) was used as the collectors, ammonium sulfate and ethylenediamine phosphate (ASEP) were used as activators. Sodium sulfide and methyl isobutyl carbinol (MIBC) were used as vulcanizing and foaming agent, respectively. Water glass was used as dispersing agent in the closed-circuit flotation process. Hydrochloric acid (HCl) and sodium hydroxide (NaOH) were used as the pH regulators. In addition, all reagents used in flotation tests were of analytical grade. Deionized distilled water was used for micro-flotation, microscopic imaging tests, and laser particle size analysis.

*2.2. Methods*

2.2.1. Micro-Flotation Tests

A XFG-1600 type (mechanical agitation) flotation machine (Hongxing mine machinery factory, Luoyang, China) with the volume of 40 mL was used in the micro-flotation tests. The mineral suspension was prepared by adding 2.0 g of a single mineral (−74 um) to 40 mL of solutions in single mineral flotation tests. The pH of the mineral suspension was adjusted to the desired operating value by adding HCl or NaOH stock solutions. Flotation concentrates were then collected for a total of 5 min. In flotation dynamics tests of malachite, the general reagent addition scheme involved

ASEP, sodium sulfide and collector ZH-1 or ISX, then MIBC frother addition with each stage having a 2-min conditioning periods prior to the addition of the next reagent. Concentrates were collected 5 min flotation time with a scraping rate of every 2 s. The following conditions were constant in all tests, unless otherwise noted: (1) a superficial gas velocity 3.11 cm/s, (2) impeller rotational speed of 1900 rpm, (3) solids concentration of 4.7%, (4) frother addition of 20 mg/L at the beginning of the test, (5) a froth depth of about 0.5 cm. In single mineral flotation, the recovery was calculated based on solid weight distribution between the two products. Product wet and dry weights were used to determine the water recovery.

### 2.2.2. Recovery by Entrainment

Particles were recovered via two primary mechanisms: (1) by true flotation, in which hydrophobic particles were recovered by attachment to air bubbles, and (2) by entrainment, which involves the nonselective recovery of particles in water to the concentrate, irrespective of whether they are hydrophobic or hydrophilic [18–20]. It is worth noting that the water in the cell used in this calculation included both the water initially added to the flotation cell as well as the distilled water which was added to maintain the pulp level. This added water was dosed with frother at the same concentration as that added at the beginning of the test (i.e., 20 mg/L) with the aim of achieving a relatively constant concentration of frother in the cell throughout each experiment. The equation to calculate water recovery is shown as follows:

$$R_w = \frac{w_1}{w_2 + w_3} \tag{1}$$

where $w_1$ is the weight of water in the foam concentrate, $w_2$ is the water initially added to the flotation cell, and $w_3$ is distilled water which was added to maintain the pulp level. Caution should be required when comparing the water recovery values presented in this paper with those presented elsewhere, as they may have been calculated differently. The recovery by entrainment was virtually the water-recovered value corrected by the degree of entrainment. The simplified model of gangue recovery by entrainment commonly seen in the literature is as follows [21]:

$$R_{ent,i} = ENT_i \cdot R_w \tag{2}$$

where $R_{ent,i}$ is the recovery by entrainment (for particle size $i$), $R_w$ is the water recovery, and $ENT_i$ is the degree of entrainment for particle size $i$. An empirical mathematical model [16] for the degree of entrainment was proposed to relate the degree of entrainment to particle size and density, as shown in Equation (3):

$$ENT_i = \frac{1}{1 + 0.061 \cdot \left[ (\rho_s - \rho_w) \cdot d_i^2 \right]^{0.689}} \tag{3}$$

where $\rho_s$ is the specific gravity of the solid particles, $\rho_w$ is the density of the water, $d_i$ is the particle diameter. It should be noted that in the experiments performed in the current study only particle size was varied, while the other operating conditions were kept constant, such as fluid viscosity and particle shape. In addition, the parameters used by the authors are not the same as those used by Wang et al. [22].

### 2.2.3. Laser Particle Size Experiments

For the particle size analysis, 2.0 g of single mineral was taken and made up to 40 mL after the addition of distilled water in 1000-mL beakers. The corresponding flotation agents, which were the same types as those used in the flotation test, were added into the pulp and stirred for 5 min. The mean size of the pulp was then measured with a laser particle size analyzer at a spindle speed of 1900 rpm (MasterSizer 2000, Malvern Panalytical Ltd., Malvern, UK).

### 2.2.4. Image Analysis

An optical microscope equipped with a digital camera was used to observe the fine particle flocculation. The sample was prepared by dropping it on a glass plate, followed by putting a thin glass slide on the sample to fix it. During the observation, the floc particles were photographed [15].

### 2.2.5. Batch Flotation

The batch flotation tests were carried out in the XFD flotation cell with a volume capacity of 1 L. It is worth to notice that the three different flotation stages (roughing/scavenging/concentrating) conducted in the different flotation machines, but the equipment model is the same with the same size of the cell. The feed samples with particle size −25 μm/+25 μm were used in tests. The flotation concentration (55%, mass percent) was adjusted to a desired level by tap water. The flotation flowsheets of the experiments were shown in Figure 2a,b.

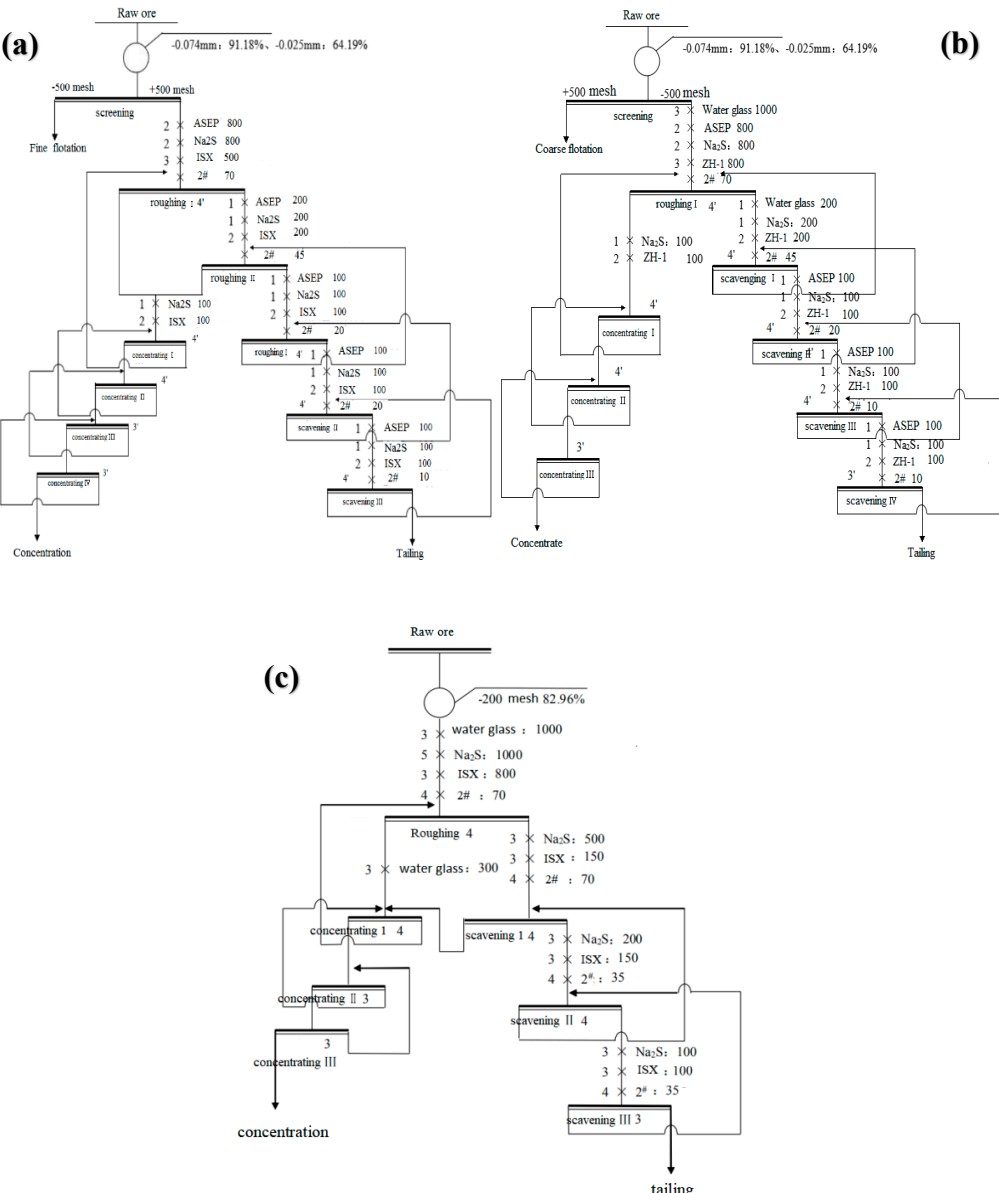

**Figure 2.** The batch flotation flowsheets of the (**a**) +25 μm; (**b**) −25 μm and (**c**) −74 μm ore (**c**) (the unit of dosage is g/t).

Besides, the flotation flowsheet of the normal experiment (without staged-flotation) was shown in Figure 2c. All of the concentrate and tailings were filtered, dried and weighed. All of the closed-circuit experiments were repeated five times, and the results were averaged respectively at the end.

## 3. Results and Discussion

### 3.1. Flotation Results

#### 3.1.1. Analysis of the Difference in Flotation Behavior

Figure 3 shows the relationship between the collector dosage and the flotation recovery of malachite and azurite. It was indicated that the collector ISX has strong collection performance for the azurite, the flotation recovery exceeded 95% when the dosage of ISX was 150 mg/L. However, the recovery of malachite was relatively lower only near 80% under the same conditions.

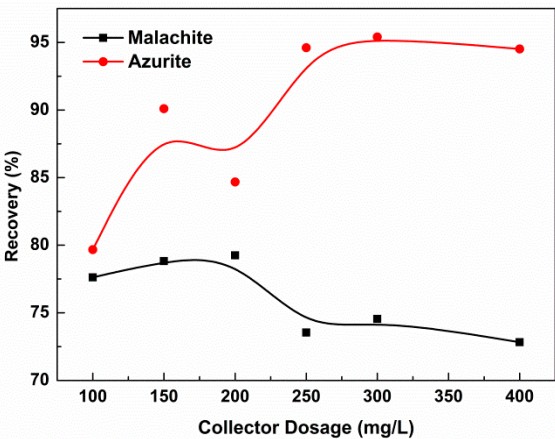

**Figure 3.** The recovery of copper oxide ore under conditions of different collector dosage (pH = 8.5).

After careful analysis, it was found that the cause of the significant difference in floatability between the two minerals may be the distribution of grain size, as shown in Figure 4.

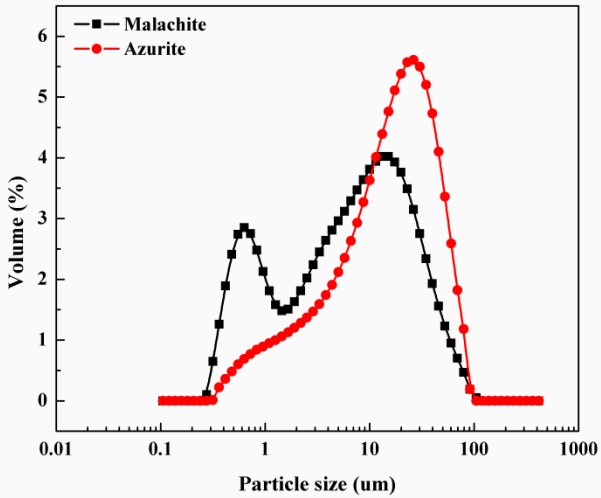

**Figure 4.** Particle size analysis results of azurite and malachite.

Obviously, the particle size distribution of malachite in the −10 μm was much higher than that of the azurite, which is one of the most likely reasons why the flotation recovery of malachite was lower than that of the azurite. Therefore, we focus on the specific particle size −10 μm in order to find

a suitable method to further improve the flotation recovery of malachite. Details are shown in the next section.

### 3.1.2. The Collecting Performance of the New Collector ZH-1

Figure 5 shows the relationship between the collector dosage and the recovery of −10 µm malachite. It was demonstrated that the recovery of malachite increases significantly as the amount of collector increases, until the dosage exceeds 150 mg/L and 250 mg/L for ISX and ZH-1, respectively. It was clear that the use of ISX alone has lost its effectiveness in flotation of fine-grained malachite. But the new type of xanthate (ZH-1, C3-5 carbon chain xanthate combination) shown a high-efficiency collection performance for fine-grained malachite. Especially, the recovery of fine-grained (−10 µm) malachite even more than 95% when the ZH-1 dosage was 150 mg/L.

However, it should be noted that the ZH-1 contains a small amount of alcohols because of the production process and purity of the product, which means it possessed a certain foaming ability [23]. The single mineral experiment, recovery rising sharply for fine-grained malachite was possibly more attribute to rich bubble inclusions rather than the collecting property as a collector. Therefore, subsequent chapters will explore this conjecture to determine whether the new collector ZH-1 is truly effective with the recovery of fine-grained malachite, or simply because of the ability like foaming agent.

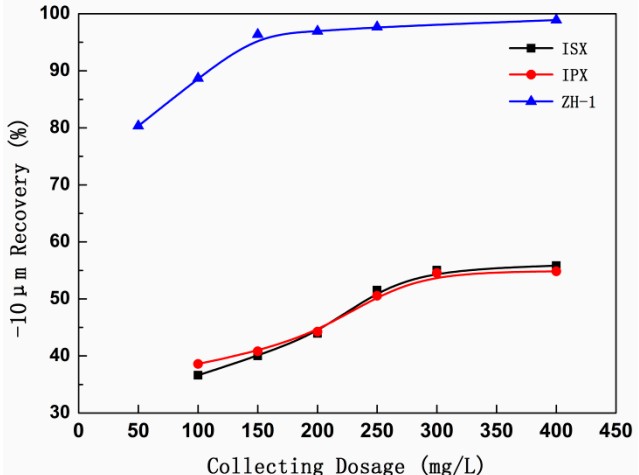

**Figure 5.** Relationship between the collector dosage and the recovery of fine-grained malachite (pH = 8.5).

### 3.2. Flotation Entrainment Rate Analysis

The collective effect of the new collector ZH-1 can be verified through the experimental results of the actual recovery and the entrainment recovery. The ASEP (150 mg/L) was used as activator, sodium sulfide (200 mg/L) as vulcanizing agent, pulp pH = 8.5 and MIBC (20 mg/L) as frother at pulp pH = 8.5 in this experiment.

In the experimental test system, the average particle size of the particles was less than 30 µm, so it can be assumed that all the malachite particles were close to being spherical particles; secondly, the viscosity of the slurry with different collector grades of malachite under the condition of adding the collector using rheometer test, the results show that the viscosity of the pulp tends to be the same under the three-grain condition, which is about 1 mPa·s, which is the viscosity of water at room temperature (20 °C). This is because the concentration of the slurry used in the single mineral flotation test is as low as about 4.7%, resulting in the viscosity of the slurry being close to the viscosity of water [16]. In summary, this experiment can be calculated by the empirical formula provided in Section 2.2.2.

Table 3 and Figure 6 show the results of the actual collection ability of different collectors for each size fraction of malachite. It can be seen from the results that the two collectors have obvious effects on the capture of two fractions of −38 + 25 µm and −25 + 10 µm malachite, and the actual recovery

($R_{Act}$: Total recovery minus the entrainment recovery $R_{ent.i}$) was increased to over 90%. Although the collection performance for both of collector ZH-1 and ISX is similar, the entrainment recovery of the three size fraction by ZH-1 was much higher than SIX. For improving the quality of concentrate in the practice production, the collector SIX should be considered priority for +25 μm malachite flotation during the staged flotation process. On the other hand, ISX has a poor performance on the collection of −10 μm malachite and the actual recovery only increased from 5% to 15%. On the contrary, the collector ZH-1 exerts a strong collective effect on −10 μm malachite, and the actual recovery was increased to about 50%.

As can be seen from the comprehensive analysis, the best collector select should be considered according to the particle size of malachite during the staged flotation process. This means the ZH-1 is beneficial to the recovery of malachite at −25 μm, but ISX shows a better collecting ability for malachite at +25 μm.

**Table 3.** The entrainment recovery of the three size fractions in the presence and absence of collector.

| Collector | Fraction D | Total Recovery% | Average Size $d_i$ (μm) | $R_w$ (%) | $R_{ent.i}$ (%) | $R_{Act}$ (%) |
|---|---|---|---|---|---|---|
| Without | −10 μm | 31.11 | 4.837 | 63.124 | 25.957 | 5.153 |
| | −25 + 10 μm | 46.89 | 18.761 | 32.232 | 3.226 | 43.664 |
| | −38 + 25 μm | 56.23 | 30.987 | 14.518 | 0.325 | 55.905 |
| With ISX | −10 μm | 49.96 | 4.921 | 72.234 | 34.535 | 15.425 |
| | −25 + 10 μm | 94.89 | 18.887 | 40.124 | 4.061 | 90.829 |
| | −38 + 25 μm | 96.78 | 30.791 | 15.232 | 0.632 | 96.148 |
| With ZH-1 | −10 μm | 90.12 | 4.881 | 75.983 | 40.939 | 49.181 |
| | −25 + 10 μm | 98.72 | 18.564 | 51.231 | 9.201 | 89.519 |
| | −38 + 25 μm | 99.81 | 30.654 | 45.324 | 4.69 | 95.120 |

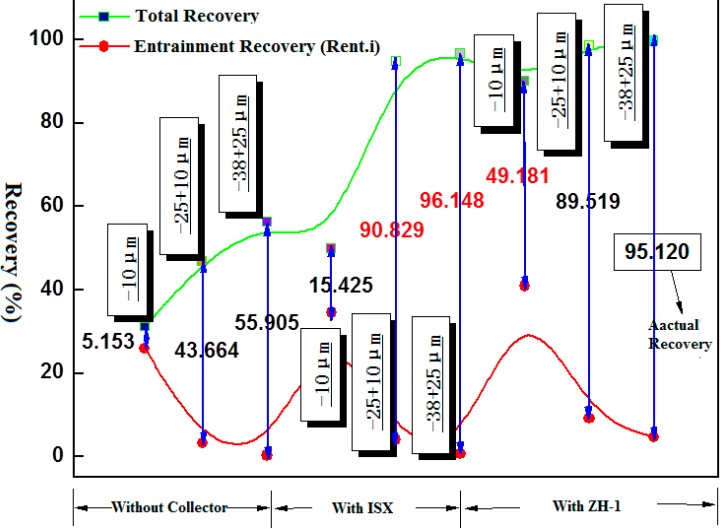

**Figure 6.** The recovery of true flotation via bubble attachment of the three size fractions in the presence and absence of collector (200 mg/L, pH = 8.5; the red marked represented the best true flotation recovery for the three fractions of malachite).

### 3.3. Laser Particle Size Analysis

Figure 7 shows the relationship between different collectors and the average particle size of malachite. It was indicated that the average particle size of −10 μm malachite was about 4.641 μm. When the ISX and the IPX were added, the average particle size of the malachite was essentially unchanged. On the other hand, the malachite particle size increased sharply to 9.631 μm when the

ZH-1as collector was used. ZH-1 has a strong flocculation ability, and the fine-grained malachite can be flocculated into a mass then increase the probability of collision with bubbles, thereby improving the recovery of the fine-grained malachite [17,24].

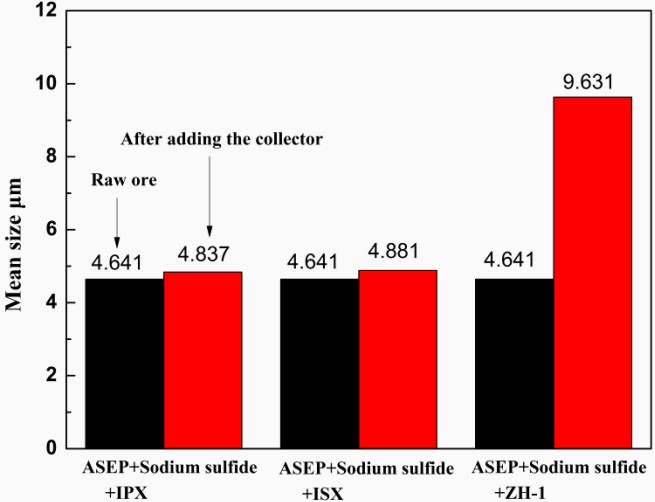

**Figure 7.** The effect of different collectors on the average particle size of malachite (−10 μm).

### 3.4. Microscopic Analysis

The flocculation of the fine-grained mineral particles after the addition of the ZH-1 collector was analyzed by electron microscopy, and the results are shown in Figure 8. According to the image of the fine particles, these appear dispersed throughout the field of view, but the fine particles condensed into a mass when the collector ZH-1 was added. In addition, Figure 8 also demonstrates that the fine-grained minerals also can be adsorbed on the surface of the larger granular minerals, it is beneficial to increase the mineral particle size and promote the adsorption of mineral particles on the surface of the bubble to achieve bubble mineralization, as shown in Figure 9.

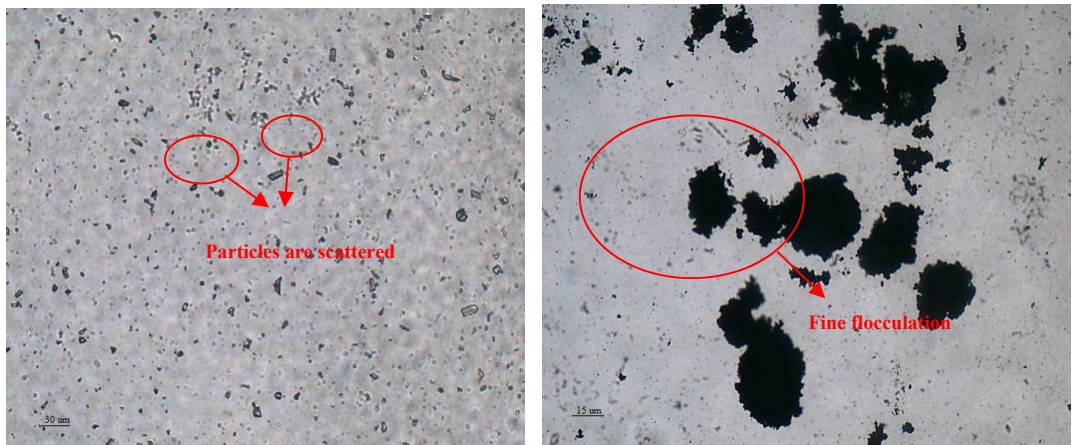

**Figure 8.** Distribution of malachite particles under the microscope before and after adding ZH-1.

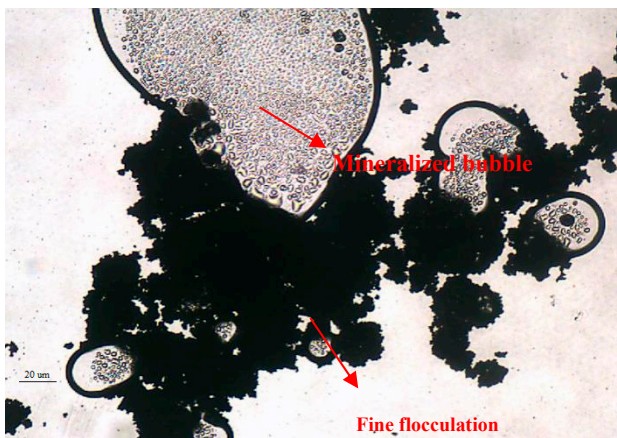

**Figure 9.** Analysis of mineralized bubble by microscope imagine.

### 3.5. Locked-Cycle Tests of Grading Flotation

The technical approach used in the Dishui Copper Processing Plant in the past was ungraded direct flotation (two roughing, three concentrating, three scavenging steps). The reagent regime was: grinding fineness −0.074 mm 93.98%, the dosage of ASEP and sodium sulfide were 800 g/t respectively, and the addition of collector ISX was 1200 g/t, the water glass was used as dispersing agent in rough selections with 1000 g/t, the perfect pulp pH value in the flotation was about 8.5–9.0. The indicators obtained by the open circuit experiment were indicated in Table 6.

#### 3.5.1. Locked-Cycle Test of +25 μm

500 g of dry ore was taken each time, then 50% grinding concentration were ground for 3 min (−0.074 mm content 91.18%, −0.025 mm content 64.19%). Grinding products were sieved by 25 μm quasi-screens. The on-screen products were selected and remove to a 1.0L flotation cell for flotation operation. The locked-cycle experiment flow sheet shown in Figure 2a, and the test results were listed in Table 4. It can be seen from the results that the collector ISX shown a good selective collection ability on +25 μm copper oxide ore, and the copper grade in the tailings was only 0.14%, and the recovery was up to 91.94% with a grade of 18.24%.

**Table 4.** Flotation grade and recovery of +25 μm malachite.

| Grading Flotation | Product | Yields /% | Cu Recovery /% | Cu Grade /% |
|---|---|---|---|---|
| | Concentrate | 8.05 | 91.94 | 18.24 |
| +25 μm | Tailing | 91.95 | 8.06 | 0.14 |
| | +25 μm | 100 | 100 | 1.6 |

#### 3.5.2. Locked-Cycle Test of −25 μm

After sedimentation of the product under the sieve (−25 μm) was clarified, the supernatant liquid was poured out as the flotation water, and the 1.5 L flotation cell is used for flotation. The locked-cycle experiment flow sheet shown in Figure 2b, and the test results were listed in Table 5. It can be seen from the results that the collector ZH-1 shown a good selective collection ability for −25 μm copper oxide ore, and the copper grade in the tailings was only 0.24%, which realizes efficient flotation recovery of fine-grained copper oxide.

**Table 5.** Flotation grade and recovery of −25 μm malachite.

| Grading Flotation | Product | Yields /% | Cu Recovery /% | Cu Grade /% |
|---|---|---|---|---|
| | Concentrate | 3.96 | 75.60 | 18.03 |
| −25 μm | Tailing | 96.04 | 24.40 | 0.24 |
| | −25 μm | 100.00 | 100 | 0.94 |

Table 6 shows the flotation grade and recovery of malachite with or without staged flotation operation. The results indicated that the copper oxide flotation recovery increased from 79.67% to 83.38%, and the grade also raised from 18.08% to 18.14% after using the staged flotation technology. The staged flotation process was quite effective for the recovery of copper oxide at the Dishui Copper Processing Plant. However, it was necessary to consider the economic feasibility since the use of staged flotation requires the transformation of existing equipment and plants.

**Table 6.** Flotation grade and recovery of malachite with or without grading flotation operation.

| Category | Product | Yields /% | Cu Recovery /% | Cu Grade /% |
|---|---|---|---|---|
| | Concentrate | 5.39 | 83.38 | 18.14 |
| With Grading Flotation | Tailing | 94.61 | 16.62 | 0.21 |
| | Raw ore | 100 | 100 | 1.17 |
| | Concentrate | 5.14 | 79.67 | 18.08 |
| Without Grading Flotation | Tailing | 94.86 | 20.33 | 0.25 |
| | Raw ore | 100 | 100 | 1.17 |

*3.6. Economic Feasibility Analysis*

(1) Analysis of new economic benefits

After expansion of the Dishui Copper Processing Plant, the annual ore dressing capacity will reach 1 million tons, and the copper grade of raw ore was about 1.0%. The annual increase in metal copper output will increase by one recovery rate:

$$1,000,000 \text{ ton} \times 1.0\% \times 1\% = 100 \text{ ton} \tag{4}$$

According to the current market copper price of 6571 US $/ton, the return rate of 82%, each increase in the recovery rate will increase the metal copper sales revenue:

$$100 \text{ ton} \times 6571 \text{ US \$/ton} \times 82\% = 657,100 \text{ US \$} \tag{5}$$

The closed-circuit test results show that the staged flotation process can increase the copper recovery by more than 3%. Increasing the recovery by 3% will increase the output of metal copper by about 300 ton per year, increase the sales revenue by about 1.9 million US$, and after deducting 17% of the value-added tax, increase the gross profit by about 1.6 million US$.

(2) Investment and cost increase analysis

To use the mud-sand sorting process, an additional set of cyclones and a flotation system are required. The flotation system can use the original 1000T mineral processing plant idle flotation equipment (the power of a single unit is 7.5 KW, a total of 18 flotation machines), the swirling device grading effect will have a greater impact on the yield of +25 μm, so it is necessary to select according to the fine characteristics of the grinding products and the requirements of fine-grain grading. The addition of a flotation system will result in an additional three flotation operators (salary about 8000 US$ for one person per year), and some input will be required for water and electricity consumption and spare parts.

## 4. Conclusions

The collector ISX has strong collection performance for azurite, and the flotation recovery exceeded 95% when the dosage of ISX was 150 mg/L. However, under the same conditions, the recovery of malachite was relatively low, only near 80%, due to the higher particle size distribution in the −10 μm than that of the azurite. Use of ISX alone has lost its effectiveness in flotation of fine-grained malachite, but a new type of xanthate (ZH-1, a $C_{3-5}$ carbon chain xanthate combination) has shown a high-efficiency collection performance for fine-grained malachite. Especially, the recovery of −10 μm malachite was higher than 95% when the ZH-1 dosage was 150 mg/L.

The two collectors (ISX and IPX) had obvious effects on the capture of two fractions of −38 + 25 μm and −25 + 10 μm malachite, and the actual recovery was basically increased to over 90%. The collector ZH-1 showed a strong collection ability for −10 μm malachite, and the actual recovery was increased to about 50%, due to its strong flocculation ability, with the average particle size of −10 μm malachite sharply increasing from 4.641 μm to 9.631 μm.

The copper oxide flotation recovery increased from 79.67% to 83.38%, and the grade also raised from 18.08% to 18.14% after using the staged flotation technology. This technology was quite effective for the recovery of copper oxide at the Dishui Copper Processing Plant, successfully increasing the gross profit by about 1.6 million US$ per year.

**Author Contributions:** R.Z., G.G., Z.C. conceived and designed the experiments; R.Z. and S.S. performed the experiments and analyzed the data; R.Z. and G.G. contributed reagents and materials; R.Z., Z.C. and Y.W. wrote the paper.

**Funding:** The authors acknowledge the support of the National Natural Science Foundation of China (No. 51374249), the National Key Technology R&D Program (No. 2015BAB12B02) and the Science and Technology Planning Project Guangdong Province, China (No. 2013B090800016).

**Conflicts of Interest:** The authors declare no conflict of interest.

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
