# Peer review of "A New Collector for Effectively Increasing Recovery in Copper Oxide Ore-Staged Flotation"

_minerals, doi:10.3390/min9100595_

Round 1

Reviewer 1 Report

The manuscript entitled "The New Method for Effectively Increasing the Recovery of Copper Oxide Ore - Staged Flotation" investigated the usage of a new collector mixture on increasing the flotation recovery of very fine malachite particles (-10 micron) as obtained for azurite with different types of collector. The presentation of introduction and the discussion of past studies were clearly presented but it is worth to note that some additional discussions would increase the novelty of this study. Although the paper can be acceptable in its current form, I think it would be better after minor revisions listed below;

-In Line 179, it was mentioned that the low recovery of malachite could be ascribed to the amount of very fine sized ranged particles. And in further sections, it was mentioned that by adding ZH-1 type collector, a considerable increase was obtained in average size of feed. It can be ascribed to that reason but I think it would be better if additional adsorption tests were carried out at least for one concentration for both new type and other well-known collectors in order to relate these findings with their adsorption data (but this is not mandatory)

- And please use only "mesh" or "μm" within whole paper for showing the dimensions of particles.  

Author Response

Dear Reviewers,

please find my responses in attached file.

Kind regards

Reviewer 2 Report

This paper is an interesting study on the use of a new collector reagent for the flotation of ultrafine particles of copper oxide ore, and in my opinion, should be published after some improvements.
The title does not represent the work: the study focus on the effect of the new collector ZH-1 not on the new method. Therefore, a better title must be included.
The entrainment recovery is determined by experiment and modeling. However, little discussion is included in the results. For example, and based on table 3, which are the best conditions? What is the trade-off between recovery by true flotation and entrainment? Which are the best operation conditions?
The flotation kinetics was mentioned in section Methods, but no results and discussion were included later in the manuscript.
Figures 8 and 9 can be improved including size scales.
Line 105, delete word "double"

Author Response

(The authors gave the same response as above.)

Reviewer 3 Report

Dear Authors,

please find my remarks in attached file.

Kind regards

Author Response

(The authors gave the same response as above.)

Round 2

Reviewer 2 Report

Please, add a column in Table 3 with true flotation recovery.

Author Response

Dear Reviewers,

        Thank you for your valuable advice. The detailed true recovery of flotation was added in  Table 3, related content was marked in red in our revised manuscript. Thanks again for your contribution to improve the quality of our work.

Authors

Reviewer 3 Report

Dear Authors,

please find my a few answers in the file.

Kind regards

Author Response

Dear reviewers,

       Thank you for your suggestion and advice to further improve the quality of our manuscript. All of your recommendations have been adsorbed and revised in our revised manuscript, which you can find in the upload system.  Besides, the main responses to your suggestions are as following:

1). The phrase " monomer dissociation" was replaced by "liberation". Indeed, the word you advised was more suitable and clearer for the reader to understand.

2). The corresponding data in Table 1 was added. All of the low concentration (basically negligible) of components in the analysis samples were combined into the Others.

3). Firstly, we should apologize for our carelessness leave out this revising suggestion. Now, the superfluous data like data about d10 and d50 have been removed from Table 2.

4). Indeed, more details about the flotation equipment used in the closed-circuit operations should be included in our manuscript. The corresponding information was added in the paper, related content was marked in red.

5). Yes, such a sentence should be emphasis on the paper. Therefore, we added the corresponding statement in the part of 2.2.6

We tried to best improve the manuscript and made some changes to the manuscript. We appreciate for Reviewers’ warm work earnestly, and hope that the correction will meet with approval.